# High-Resolution Permanent Magnet Drive Using Separated Observers for Acceleration Estimation and Control [note 1]

**DOI:** 10.3390/s22030725

**Published:** 2022-01-18

**Authors:** Yi-Jen Lin, Po-Huan Chou, Shih-Chin Yang

**Affiliations:** 1Department of Mechanical Engineering, National Taiwan University, No. 1, Sec. 4, Roosevelt Rd., Taipei 10617, Taiwan; f08522816@ntu.edu.tw; 2Department of Advanced Servo Technology, Industrial Technology Research Institute, No. 195, Sec. 4, Chung Hsing Rd., Chutung, Hsinchu 310401, Taiwan; phchou@itri.org.tw

**Keywords:** PM motor system, disturbance rejection, acceleration control and estimation

## Abstract

This paper proposes a high-resolution permanent magnet (PM) motor drive based on acceleration estimation and control. The PM motor is widely implemented in the printed circuit board (PCB) manufacturing process. To achieve the demanded 1 μm drilling resolution, a sine/cosine incremental encoder is usually installed for motion control. In this paper, several improvements are developed to increase the motion control steady-state accuracy balancing transient response. First, the interpolation of every two encoder counts is proposed to increase the position sensing resolution. In this case, the transient response is improved through the high-resolution position feedback. Second, a closed-loop observer with two independent bandwidths is proposed for acceleration estimation. By using the interpolated position for acceleration estimation, the vibration-reflected high-frequency torque harmonics can be compensated through the acceleration closed-loop control. It reduces the steady-state error under the same sensing hardware. According to experimental results, both transient response and steady-state error can be improved on a PM motor using the proposed position interpolation and acceleration control.

## 1. Introduction

The PM motor combined with laser drilling has been widely implemented in PCBs. For modern PCB manufacturers, laser drilling, instead of mechanical drilling, is preferred for PCB hole digging because of its high response and high accuracy [1]. According to the commercial product from Hitachi Via Mechanics [2], laser drilling accuracy can be within 7–15 μm. To achieve comparable performance within a demanded response, PM motors or voice coil motors are used for the control of laser mirrors. The overall laser drilling system with PM motor is illustrated in Figure 1.

Due to the power limitation on voice coil motors, the laser drilling system with PM motors is preferred for PCB hole drilling [3]. Considering the micrometer-level digging accuracy, high-resolution position sensors are required for motion control. In [4], a sine/cosine encoder is installed in a PM motor. The position measurement resolution is improved by compensating both signal offset and magnitude deviation. In [5,6], the resolution of the same analog encoder is enhanced by the digital phase-locked loop with the phase and frequency transformation. However, these improvement methods are based on the signal condition correction. The position resolution is still limited below the maximum available encoder pulses. In [7], the electronic interpolation is added between two encoder counts. In this case, the position resolution can be increased by adding the ADC resolution into total quadrature pulses. However, the ADC sampling delay at high speed degrades this interpolation process [8]. Until now, this sensing modulator with signal gain and offset compensation has been implemented in a commercial IC, e.g., Texas Instruments TIDA-00176 [9].

High-bandwidth motion control is also key for PM motor systems. Due to the bandwidth requirement on current regulation, the hardware implementation using analog amplifier circuits is preferred [10,11]. For motion control implementation, computer-based position control can be designed to cascade with this current regulation hardware for the mirror adjustment. In order to increase the position control response, the feedforward control is added based on this physical model [12,13]. It is noted that the speed signal is required to formulate the feedforward controller. Because the speed is obtained from the differentiation of measured position, the speed observer is developed for speed estimation without differentiation noises [14,15]. However, the observer estimation accuracy is strongly dependent on the PM motor model and parameter accuracy [16,17].

Considering the laser drilling at high speed, motor vibration is observed due to the unbalanced mirror [18]. In this case, vibration load estimation and compensation can be applied to improve the mirror-reflected vibration [19,20]. In [21], the vibration load is predicted based on the mechanical model of mirror mass, spring, and damper system. However, the load compensation is highly dependent on the parameter accuracy. It is noted that the disturbance observer is also used to estimate the disturbance torque for the vibrational load compensation [22]. However, different from prior speed observers [14,15], the torque feedforward cannot be implemented in the disturbance observer, because no disturbance information is included in the feedforward command [23]. Without the torque feedforward, the disturbance torque estimation and rejection are only useful at a low frequency region. By contrast, the vibrational load can be estimated with the knowledge of motor acceleration, because the load variation affects the acceleration condition [24]. However, the accelerometer must be installed for vibration load compensation [25]. Since the motor is inside a closed environment with high temperature, this accelerometer installation is a challenge for the PM motor [26]. In [27], the acceleration is obtained from the acceleration observer reconstructed by the measured position. Different from the disturbance observer, the acceleration observer contains the torque feedforward to estimate the motor instantaneous acceleration. With the torque feedforward, the acceleration information at high frequency can be obtained. By applying the estimated acceleration for feedback control, the disturbance torque at high frequency can be compensated [23]. Nevertheless, considerable acceleration estimation noises are observed, because the acceleration is estimated by two times of position differentiation, though the accelerometer is removed.

This paper proposes a high-performance motion control for the PM motor to improve the vibration torque harmonic through the encoder position interpolation, speed estimation, and acceleration estimation/control. The proposed motion control is extended from [28] with comprehensive analysis on the encoder position interpolation, speed estimation, and acceleration estimation/control. Although acceleration estimation was originally developed in [27], considerable estimation noises with degraded steady-state errors are resultant due to two times of position differentiation. In this paper, several improvements are developed to increase the motion control accuracy balancing transient response. The encoder pulse interpolation is firstly developed using a sine/cosine encoder. The overall sensing resolution is increased from the interpolation of sampled sine and cosine position signals. After obtaining the high-resolution position, separated observers with different bandwidths are proposed for the acceleration estimation. The first observer estimates the speed with low bandwidth to remove differentiation noises. The second observer combines the estimated speed and interpolated position for high-bandwidth acceleration estimation. By using the acceleration control, the vibration reflected torque harmonics at high frequency are reduced, while the transient response is maintained. It is concluded that the motion control resolution can increase to 1 μm by combining the proposed position interpolation and acceleration control. The proposed high-performance disturbance rejection is implemented in a PM motor for experimental validation.

## 2. PM Motor System

This section explains the PM motor analyzed in this paper. Figure 2 shows the test PM motor topology. In this motor, four magnets are mounted on the rotor. Coreless windings are designed in the stator for the electromagnetic torque production. The test motor is based on the prototype of Canon GM-1010 for the investigation. By using the existing drive provided for GM-1010, the control accuracy is up to 4.41 × 10^−5^ deg, as reported in the datasheet [29]. More importantly, the resolution is expected to improve to 1.08 × 10^−5^ deg using the proposed motion control method. Detailed PM motor specifications are listed in Table 1.

Figure 3 explains the PM motor system. In this system, the motor current and position are measured for motion control and current control. Considering the application in high speed PCB drilling, the current regulation bandwidth needs to achieve several kilohertz [12]. In this case, the controller through hardware implementation is preferred without the digital calculation delay. Considering the difference between the electrical and mechanical systems, the position control bandwidth can be designed at one tenth (1/10) of current regulation bandwidth. For the implementation of advanced control algorithms, digital motion control is preferred, especially for vibration torque estimation and compensation.

## 3. High-Resolution Position Sensing

This section explains the position measurement through the sine/cosine encoder sensor. In this paper, a magnetic encoder with 512 pulses is installed for the closed-loop motion control. In order to increase the position measurement resolution, the position interpolation is proposed through ADC sample of sine/cosine encoder signals.

The corresponding position interpolation is illustrated in Figure 4. Considering firstly the encoder pulse accumulation, two comparators are used to obtain quadrature pulses for the coarse position measurement. The total pulse resolution is 9-bit. After that, two ADC channels are used to calculate the fine position based on a mathematical algorithm. On that basis, the encoder sine/cosine continuous signals are sampled through 16-bit ADC. An arctangent calculation is applied for the interpolation between every two-position count. Ideally, the position resolution is increased to 25-bit, combining 9-bit pulses and 16-bit ADC. Figure 5 shows the position interpolation comparison with different position signals. In (a), the conceptual waveform is given, where the fine position $\theta_{\mathrm{fine}}$ is interpolated every two position counts. The actual test position waveform is compared in (b) from the test PM motor. For the coarse position $\theta_{\mathrm{crs}}$, the position resolution is around 360 deg/2^9^ = 0.7 deg. As listed in Table 1, the corresponding scanner resolution is 78 μm. By using the interpolated position $\theta_{\mathrm{intp}}$, the position resolution is increased to 360 deg/2^9+16^ = 1.08 × 10^−5^ deg. The scanner resolution is theoretically below 1 μm, assuming a rigid body on the scanner mirror and motor shaft. This increased position resolution is useful for acceleration estimation and acceleration control.

## 4. Dynamic Model of PM Motor

This part explains the dynamic model of the PM motor system. Figure 6 shows the completed model for the PM motor with mirror analyzed in this paper. In this figure, $J_{M}$ and $J_{ML}$ are, respectively, the motor and mirror inertia; B_ML_ and K_ML_ are equivalent damper and spring caused by mirror load; and B_M_ and K_M_ are damper and spring considering the motor rotation dynamic. Under this effect, the corresponding torque load can be modelled by
(1)$$T_{L}(t)=B_{\mathrm{ML}}\frac{d}{dt}[\theta_{M}(t)-\theta_{\mathrm{ML}}(t)]+K_{\mathrm{ML}}[\theta_{M}(t)-\theta_{\mathrm{ML}}(t)]+B_{M}\frac{d\theta_{M}(t)}{dt}+K_{M}\theta_{M}(t)+T_{\mathrm{dis}}(t)$$
where θ_ML_ and θ_M_ are the position, respectively, of the mirror and motor, and $T_{\mathrm{dis}}$ is used to model the vibration torque induced by mirror unbalance load. Based on the above derivation, the PM motor model is formulated in S-domain to easily analyze the dynamic property. The model input is actual current I and the torque load T_L_, where the output is position θ_M_. The transfer function of θ_M_ is developed by
(2)$$\begin{array}{c} \theta_{M}(s)=\frac{K_{t}\cdot I(s)}{(J_{M}+J_{ML})s^{2}}-\frac{T_{L}(s)}{(J_{M}+J_{ML})s^{2}}\\ \theta_{M}(s)=\frac{T_{\mathrm{em}}(s)}{(J_{M}+J_{ML})s^{2}}-\frac{T_{L}(s)}{(J_{M}+J_{ML})s^{2}} \end{array}$$
where s = σ + jω is a complex variable with real number σ and ω in S-domain, and K_t_ is the torque constant of the PM motor. As reported in [30,31], the motor electromagnetic torque is equal to T_em_(s) = K_t_ × I(s). Figure 7 explains the conventional motion control for the PM motor with mirror load. In this system, the cascaded position and speed controller are developed for the manipulation of current command. In this controller, the position controller is formulated by proportional $K_{p1}$ control, while the speed controller is based on proportional $K_{p2}$ and integral $K_{i2}$ control. By selecting controller gains $K_{p1}$/$K_{p2}$/$K_{i2}$, the over-damped transient dynamic without overshoot can be achieved.

Moreover, the current amplifier in Figure 7 is implemented based on the hardware circuit for high-bandwidth current regulation. Because the ideal current regulation is assumed, the current digital command I* output from the motion controller is almost equal to actual current I. In addition, a torque constant $K_{t}$ is included to model the relationship between motor electromagnetic torque $T_{\mathrm{em}}$ and drive current. It is given by
(3)$$T_{\mathrm{em}}(s)=K_{t}I(s)\approx K_{t}I^{*}(s)$$
where $T_{\mathrm{em}}$ can be manipulated based on the current command I*. As mentioned in Equation (2), the torque load $T_{L}$ caused by spring and damper affects the overall motion control performance. These non-ideal mechanical attributes are also included in the overall motion control system of the PM motor shown in Figure 7. It is important that the accurate position and speed response can be achieved if the mirror vibration effect is to be compensated by the manipulated torque output $T_{\mathrm{em}}$. In the following two sections, the observer-based acceleration estimation and acceleration control will be proposed to improve the motion control performance.

## 5. Observer-Based Speed Estimation

Motion control requires the position and speed feedback for the closed-loop regulation. In general, conventional speed estimation is based on the single degree-of-freedom, where the direct position differentiation is applied. In this case, low-pass filter (LPF) may be used to remove differentiation noises. It is well known that the LPF design is the tradeoff between signal noises and phase delay. It is not suited for high-dynamic PM motor operation.

### 5.1. Proposed Speed Observer

To remove the differentiation noise without phase delay, the observer-based speed estimation with two degrees-of-freedom is proposed. Figure 8 illustrates the proposed first observer for speed estimation based on the model of the PM motor. On this basis, the speed estimation follows the Luenberger observer topology [32].

The observer inputs use both encoder position θ_M_ and actual motor current I, where the output is estimated speed $\hat{\omega }$. The corresponding transfer function for $\hat{\omega }$(s) estimation can be then depicted by
(4)$$\begin{array}{l} \hat{\omega }(s)=\left[\frac{K_{1}s^{2}{+K}_{2}{s+K}_{3}}{{(\hat{J}}_{M}{+\hat{J}}_{\mathrm{ML}}{)s}^{2}}\right]\left(\theta_{M}\left(s)-\hat{\theta }(s\right)\right)+\frac{1}{{(\hat{J}}_{M}{+\hat{J}}_{\mathrm{ML}}{)s}^{2}}{\hat{T}}_{\mathrm{em}}\left(s\right)\\ \hat{\omega }\;(s)=\left[\frac{K_{1}s^{2}{+K}_{2}{s+K}_{3}}{{\hat{J}}_{\mathrm{ALL}}s^{2}}\right]\frac{\left(\omega_{M}\left(s)-\hat{\omega }(s\right)\right)}{s}+\frac{1}{{\hat{J}}_{\mathrm{ALL}}s^{2}}{\hat{T}}_{\mathrm{em}}\left(s\right) \end{array}$$
where ${\hat{J}}_{\mathrm{ALL}}$ = ${\hat{J}}_{\mathrm{ML}}$+${\hat{J}}_{M}$ is the estimated total inertia, and K_1_, K_2_, and K_3_ are observer controller gains for the estimation bandwidth determination. In addition, the estimated motor torque ${\hat{T}}_{\mathrm{em}}$ is designed as torque feedforward for the speed observer. In (4), ${\hat{T}}_{\mathrm{em}}$ can be obtained through the estimated torque constant ${\hat{K}}_{t}$ times the actual motor current I, as given by
(5)$${\hat{T}}_{\mathrm{em}}(s)={\hat{K}}_{t}I(s)$$
where the relationship between motor electromagnetic torque $T_{\mathrm{em}}$ and drive current I has been proved by [30,31]. As seen in Figure 8, the actual current I after current regulation is selected in Equation (5) to calculate the motor torque ${\hat{T}}_{\mathrm{em}}$. Assuming the ideal parameter estimation ${\hat{K}}_{t}$ = $K_{t}$, the estimated ${\hat{T}}_{\mathrm{em}}$ is equivalent to $T_{\mathrm{em}}$. In general, $T_{\mathrm{em}}$ should be attributed to
(6)$$T_{\mathrm{em}}(s)=K_{t}I(s)=J_{\mathrm{ALL}}\alpha_{M}(s)+T_{L}(s)$$

Considering the PM motor with vibrational torque harmonics, ${\hat{T}}_{\mathrm{em}}$ calculated by the actual current I is selected to contain the torque load $T_{L}$ information. By contrast, the current command I* can be selected instead of actual I once visible measurement noises occur on the motor system. Based on the proposed speed observer in Figure 8, the speed estimation is reconstructed by both the motor position θ_M_ and actual motor current I. The corresponding transfer function of $\hat{\omega }$ can be derived by
(7)$$\begin{array}{l} \hat{\omega }(s)={\hat{\omega }}_{\mathrm{fk}}(s)+{\hat{\omega }}_{\mathrm{ff}}(s)\\ \hat{\omega }(s)=\left[\frac{K_{1}s^{2}+K_{2}s+K_{3}}{{\hat{J}}_{\mathrm{ALL}}s^{3}+K_{1}s^{2}+K_{2}s+K_{3}}\right]\omega_{M}(s)+\left[\frac{\left({\hat{K}}_{t}/K_{t}\right)J_{\mathrm{ALL}}s^{3}}{{\hat{J}}_{\mathrm{ALL}}s^{3}+K_{1}s^{2}+K_{2}s+K_{3}}\right]\omega_{M}(s)\\ \hat{\omega }(s)=\left[\frac{\left({\hat{K}}_{t}/K_{t}\right)J_{\mathrm{ALL}}s^{3}+K_{1}s^{2}+K_{2}s+K_{3}}{{\hat{J}}_{\mathrm{ALL}}s^{3}+K_{1}s^{2}+K_{2}s+K_{3}}\right]\omega_{M}(s) \end{array}$$

In Equation (7), $\hat{\omega }$ consists of two components, feedback estimated speed ${\hat{\omega }}_{\mathrm{fk}}$ and feedforward estimated ${\hat{\omega }}_{\mathrm{ff}}$. The first term ${\hat{\omega }}_{\mathrm{fk}}$, on the left-hand side of the equation, is reconstructed by the observer feedback control. The second term ${\hat{\omega }}_{\mathrm{ff}}$, on the right-hand side of the equation, is reconstructed by the observer feedforward control. The overall relationship of ${\hat{\omega }}_{\mathrm{fk}}\left(s)/\omega_{M}(s\right)$ is similar to single degree-of-freedom estimation with a LPF. Moreover, the second term ${\hat{\omega }}_{\mathrm{ff}}$ is calculated by the torque feedforward, where the ratio of ${\hat{\omega }}_{\mathrm{ff}}$(s)/$\omega_{M}(s)$ is equivalent to a high-pass filter (HPF). At low frequency, ${\hat{\omega }}_{\mathrm{ff}}$(s)/$\omega_{M}(s){|}_{s\to 0}$, where ${\hat{\omega }}_{\mathrm{ff}}$ does not provide the estimation information to $\hat{\omega }$. By contrast, at high frequency, ${\hat{\omega }}_{\mathrm{ff}}(s)/\omega_{M}(s){|}_{s\to \infty }$ can be assumed as ${\hat{K}}_{t}$$J_{\mathrm{ALL}}$/$K_{t}$${\hat{J}}_{\mathrm{ALL}}$. Although the parameter sensitivity is resultant on $J_{\mathrm{ALL}}$ and $K_{t}$, two key advantages are gained by combining two speed signals, ${\hat{\omega }}_{\mathrm{fk}}$ and ${\hat{\omega }}_{\mathrm{ff}}$. First, $\hat{\omega }$ estimation performance is parameter insensitive and shows no phase delay within observer bandwidth. More importantly, even beyond the observer bandwidth, $\hat{\omega }$ estimation dependent on ${\hat{\omega }}_{\mathrm{ff}}$ still shows no phase delay, though the parameter sensitivity appears. By combing both ${\hat{\omega }}_{\mathrm{fk}}$ and ${\hat{\omega }}_{\mathrm{ff}}$ for $\hat{\omega }$ estimation, differentiation noise can be removed in ${\hat{\omega }}_{\mathrm{fk}}$, while the filter delay is resolved by including feedforward ${\hat{\omega }}_{\mathrm{ff}}$. Figure 9 verifies different frequency responses with respect to ${\hat{\omega }}_{\mathrm{fk}}$(s)/$\omega_{M}(s)$, ${\hat{\omega }}_{\mathrm{ff}}$(s)/$\omega_{M}(s)$, and $\hat{\omega }$(s)/$\omega_{M}(s)$. From the magnitude plot, the absolute magnitude, instead of decibel, is selected as the vertical axis unit to clearly illustrate the magnitude deviation at different frequencies. By combining feedback ${\hat{\omega }}_{\mathrm{fk}}$(s) and feedforward ${\hat{\omega }}_{\mathrm{ff}}$(s) for the proposed speed estimation, the resulting $\hat{\omega }$(s) is the same as $\omega_{M}(s)$ at all frequencies under ideal parameter estimation. From the phase plot, the speed estimation noises can be removed in ${\hat{\omega }}_{\mathrm{fk}}$(s), while the LPF reflected phase delay can be compensated by the feedforward ${\hat{\omega }}_{\mathrm{ff}}$(s).

### 5.2. Differentiation Noises Elimination

This section explains the elimination of speed estimation noises in the proposed speed observer. Considering the encoder-based motion control, the speed is traditionally obtained based on the position differentiation, as shown by
(8)$${\hat{\omega }}_{\mathrm{direct}}(s)=s\theta_{\mathrm{ns}}(s)$$

In Equation (8), ${\hat{\omega }}_{\mathrm{direct}}$ denotes the speed estimation directly from the position differentiation. If the position measurement contains high frequency noises, e.g., EMI, these noises greatly increase on ${\hat{\omega }}_{\mathrm{direct}}$ due to the differentiation calculation. Figure 10 illustrates the frequency response of ${\hat{\omega }}_{\mathrm{direct}}$(s)/θ_ns_(s) by the blue dashed line. The magnitude of |${\hat{\omega }}_{\mathrm{direct}}$(s)/θ_ns_(s)| increases with frequency noises. This speed estimation might not be suited for PM motors for high-bandwidth operation.

It is noteworthy that these high frequency noises can be reduced through the speed observer in Figure 8. Considering the influence of position noises $\theta_{\mathrm{ns}}$ on estimated position $\hat{\omega }$, the corresponding transfer function between $\hat{\omega }$ and $\theta_{\mathrm{ns}}$ can be depicted by Equation (9) based on the observer topology in Figure 8.
(9)$$\hat{\omega }(s)=\left[\frac{K_{1}s^{3}+K_{2}s^{2}+K_{3}s}{{\hat{J}}_{\mathrm{ALL}}s^{3}+K_{1}s^{2}+K_{2}s+K_{3}}\right]\theta_{\mathrm{ns}}(s)$$

Figure 10 compares the frequency plot of $\hat{\omega }$(s)/θ_ns_(s) in the speed observer and ${\hat{\omega }}_{\mathrm{direct}}$(s)/θ_ns_(s) through Equation (8). Considering the proposed observer for speed estimation, $\hat{\omega }$(s)/θ_ns_(s) is equivalent to a HPF. Once high-frequency noises occur on measured position, the influence of noises $\theta_{\mathrm{ns}}$ on $\hat{\omega }$ maintain a constant magnitude below the observer estimation frequency.

It is important that $\theta_{\mathrm{ns}}$ noises can be further decreased based on the enhanced speed observer estimation. As seen in Figure 8, the motor speed can be estimated both from conventional $\hat{\omega }$ and enhanced ${\hat{\omega }}_{\mathrm{en}}$. Different from $\hat{\omega }$(s)/θ_ns_(s) in Equation (9), only observer proportional gain K_2_ and integral gain K_3_ are used for ${\hat{\omega }}_{\mathrm{en}}$ estimation. Thus, the transfer function of ${\hat{\omega }}_{\mathrm{en}}$(s)/θ_ns_(s) is derived by
(10)$${\hat{\omega }}_{\mathrm{en}}(s)=\left[\frac{K_{2}s^{2}+K_{3}s}{{\hat{J}}_{\mathrm{ALL}}s^{3}+K_{1}s^{2}+K_{2}s+K_{3}}\right]\theta_{\mathrm{ns}}(s)$$

For the comparison between Equations (9) and (10), the characteristic equations are the same third-order dynamic systems. The corresponding eigenvalues can be determined based on observer gains K_1_, K_2_^,^ and K_3_ to ensure the closed-loop stability. Although the stability is the same between $\hat{\omega }$(s) and ${\hat{\omega }}_{\mathrm{en}}$(s), the dynamic property is different, due to there being no differential gain K_1_ in the numerator of Equation (10). Without the differential operator, both the high-frequency dynamic response and noises are decreased. Figure 10 illustrates the frequency response of ${\hat{\omega }}_{\mathrm{en}}$(s)/θ_ns_(s) as the green dashed line. Comparing to $\hat{\omega }$(s)/θ_ns_(s), the influence of position noises is decreased beyond the speed observer bandwidth. In this paper, the observer bandwidth is designed at 2.5 kHz to ensure the sufficient speed estimation dynamic for PM motor motion control. Considering the mirror-reflected torque harmonics beyond 3 kHz, these disturbances can be compensated based on another proposed acceleration observer in Section 6.

### 5.3. Simulation Result

This part demonstrates the simulation results of different speed estimations. Figure 11 compares time-domain signals of ${\hat{\omega }}_{\mathrm{direct}}$, $\hat{\omega }$, and ${\hat{\omega }}_{\mathrm{en}}$ estimation under a sinusoidal position feedback. In this simulation, a 0.002% white noise is included in the position. The proposed speed observer in Figure 8 with the estimation bandwidth at 2.5 kHz is used to obtain both $\hat{\omega }$ and ${\hat{\omega }}_{\mathrm{en}}$. It is shown that the ${\hat{\omega }}_{\mathrm{en}}$ estimation based on Equation (10) results in the lowest speed noises. Detailed improvement on the PM motor system will be shown in the experiment section.

## 6. Acceleration Estimation and Control

This section explains a high-performance motion control through the proposed closed-loop acceleration control. Considering the PM motor at high speed, the considerable torque load in Equation (2) including the mirror vibration-reflected disturbance torque $T_{\mathrm{dis}}$ significantly increases. It is noted that this high frequency vibration load can be minimized through the proposed acceleration estimation and control.

### 6.1. Acceleration Feedback Control

As seen in Figure 7, a conventional motion controller is realized through the position and speed feedback regulation. The influence of vibration torque load is compensated based on the design of position controller proportional (P) gain $K_{p1}$ and speed controller proportional and integral (PI) gains $K_{p2}$ and $K_{i2}$. In order to evaluate the disturbance rejection under different controllers, the disturbance rejection function (DRF) is defined in this paper. Considering the PM motor, the primary disturbance is modelled by $T_{L}$ in Equation (2). Under this effect, the DRF of the PM motor is defined by the ratio of load-reflected position response θ_L_ and torque load $T_{L}$. Based on the controller topology in Figure 7, the corresponding DRF is represented by
(11)$$\left|\frac{\theta_{L}\left(s\right)}{T_{L}\left(s\right)}\right|=\left|\frac{s}{J_{\mathrm{ALL}}s^{3}+\left(K_{p2}+B\right)s^{2}{+(K}_{p1}K_{p2}{+K}_{i2}{)s+K}_{p1}K_{i2}}\right|$$

As seen in Equation (11), the controller disturbance rejection performance is increased with the decrease in DRF magnitude. Figure 12 analyzes the corresponding DRF at different frequencies based on Equation (11). In this calculation, the motion controller bandwidth is designed at 1 kHz. It is observed that the mirror-reflected unbalanced torque $T_{L}$ can be minimized at low frequency (below 1 kHz) based on the adjustment of $K_{p1}$, $K_{p2}$, and $K_{i2}$. By contrast, $T_{L}$ at high frequency (beyond 1 kHz) can only be compensated by increasing either the motor inertia J_M_ or mirror inertia J_ML_. It is noted that conventional PM motors result in small inertia for high dynamic response. More importantly, due to high rotation speed, the harmonics of mirror unbalanced load can be up to 3–5 kHz. It leads to various challenges under the conventional control topology in Figure 7.

At this part, the acceleration feedback control is proposed to improve disturbance rejection at high frequency for the PM motor. Figure 13 illustrates the proposed motion control with additional acceleration control loop. In this figure, θ*, ω* and α* are, respectively, the position, speed and acceleration command. An additional proportional gain $K_{p3}$ is used to adjust the acceleration control bandwidth. By adding the proposed control, the overall DRF is modified by
(12)$$\left|\frac{\theta_{L}\left(s\right)}{T_{L}\left(s\right)}\right|=\left|\frac{s}{\left(K_{p3}{+J}_{\mathrm{ALL}}\right)s^{3}+\left(K_{p2}+B\right)s^{2}{+(K}_{p1}K_{p2}{+K}_{i2}{)s+K}_{p1}K_{i2}}\right|$$

Comparing the DRF between Equations (11) and (12), the rejection performance is improved at high frequency by adding the acceleration control. Figure 12 also compares the DRF based on the proposed acceleration control. Because the torque load directly affects the acceleration response, the disturbance rejection is increased due to an additional acceleration gain $K_{p3}$, which is equivalent to the motor inertia, as seen from the denominator in Equation (12). It is concluded that at high frequency, the scanner disturbance rejection performance can be increased under the same motion control bandwidth.

### 6.2. Acceleration Estimation

It is noted that the measurement of instantaneous acceleration is not an easy task in motion control systems. In general, a three-axis accelerometer can be attached on the motor surface for acceleration sensing [33]. However, the PM motor requires high dynamic response for fast mirror rotation. It is a challenge for the three-axis accelerometer installed on the tiny PM motor.

At this part, the encoder-based acceleration estimation, instead of an accelerometer, is proposed to find the acceleration signal. Because the acceleration is obtained by two times of position differentiation, considerable noises must appear if the observer contains the differential controller. For the enhanced speed observer in Figure 8, the enhanced acceleration ${\hat{\alpha }}_{\mathrm{en}}$ can be obtained similarly to the enhanced speed, as given by
(13)$${\hat{\alpha }}_{\mathrm{en}}(s)=s{\hat{\omega }}_{\mathrm{en}}(s)=\left[\frac{K_{2}s^{3}+K_{3}s^{2}}{{\hat{J}}_{\mathrm{ALL}}s^{3}+K_{1}s^{2}+K_{2}s+K_{3}}\right]\theta (s)$$

However, as mentioned in Figure 10, the degraded dynamic response at high frequency is resultant without the differential gain K_1_. Figure 14 shows the signal process of the separated acceleration observer. To minimize the differentiation noise balancing the high-frequency dynamic, a separated observer without the differential controller is the candidate. In this acceleration observer, the speed regulation is designed instead of the differential controller. On this basis, the measured position and estimated enhanced speed are both used as observer inputs, while the motor torque estimation in Equation (5) is designed as the feedforward control for better estimation bandwidth. In this case, the corresponding transfer function of estimated acceleration $\hat{\alpha }$ without differentiation noises is represented by
(14)$$\begin{array}{l} \hat{\alpha }(s)=\left[\frac{K_{5}s^{3}+K_{6}s^{2}}{{\hat{J}}_{\mathrm{ALL}}s^{3}+K_{4}s^{2}+K_{5}s+K_{6}}\right]\theta (s)+\left[\frac{K_{4}s^{3}}{{\hat{J}}_{\mathrm{ALL}}s^{3}+K_{4}s^{2}+K_{5}s+K_{6}}\right]{\hat{\omega }}_{\mathrm{en}}(s)\\ \hat{\alpha }(s)=\left[\frac{K_{4}s^{4}+K_{5}s^{3}+K_{6}s^{2}}{{\hat{J}}_{\mathrm{ALL}}s^{3}+K_{4}s^{2}+K_{5}s+K_{6}}\right]\theta (s)\;\left(\mathrm{Note}:{\hat{\omega }}_{\mathrm{en}}(s)=s\theta (s)\mathrm{for}\;\mathrm{simplicity}\right) \end{array}$$

It is noteworthy that the acceleration estimation bandwidth must be sufficiently high to include the information of high-frequency torque harmonics. Different from the enhanced speed observer estimation with noise consideration, this separated acceleration observer is designed with high estimation bandwidth. Considering the disturbance rejection analyzed in Equation (12) and Figure 12, the ideal acceleration feedback is assumed for simplicity. In Equation (14), however, the acceleration is reconstructed from the position feedback through the acceleration observer. As a result, the proposed disturbance rejection based on acceleration control can be degraded. In this paper, the acceleration estimation bandwidth is designed to be 5–10 times greater with respect to the position control bandwidth for the purposes of removing high frequency vibration harmonics.

### 6.3. Simulation Results

This part illustrates the simulation results of proposed observer-based acceleration estimation. Figure 15 shows the simulation of acceleration comparison among direct speed differentiation ${\hat{\alpha }}_{\mathrm{direct}}$, enhanced acceleration estimation ${\hat{\alpha }}_{\mathrm{en}}$, and separated acceleration estimation $\hat{\alpha }$. In this simulation, a 100 Hz sinusoidal acceleration command with 3.5 kHz vibration-reflected harmonic is applied. Similar to the speed simulation in Figure 11, a 0.002% white noise is also included in order to evaluate the differentiation noises. Considering the suppression of high-frequency vibrational harmonic, the acceleration observer with 5 kHz estimation bandwidth is implemented. Moreover, the speed observer bandwidth is designed at 2.5 kHz in order to maintain the motion control accuracy without high frequency vibrational harmonic. As seen from ${\hat{\alpha }}_{\mathrm{direct}}$, considerable differentiation noises appeared. By contrast, for enhanced estimation ${\hat{\alpha }}_{\mathrm{en}}$, the 3.5 kHz harmonic information disappears, which cannot be implemented for the high-frequency vibrational harmonic rejection. Based on this simulation, the separated acceleration estimation $\hat{\alpha }$ with another independent observer bandwidth is useful to maintain vibrational harmonic information without differentiation noises. The proposed $\hat{\alpha }$ is suited for acceleration feedback control and disturbance rejection.

## 7. Experimental Results

This section explains the experimental results for the PM motor using the proposed separated observers for speed/acceleration estimation and acceleration control. Figure 16 shows the photograph of the PM motor test setup. A PM motor is coupled by a laser mirror with 0.64 g·cm^2^ inertia to stimulate the high-frequency vibrational load. Table 1 in Section 2 lists key characteristics of the test PM motor. A sine/cosine encoder with 512 pulses is installed for the position measurement. Figure 3 illustrates the overall control flowchart of the PM motor. The current regulation is implemented based on the amplifier hardware. All other motion control and observer-based estimation is implemented in a 32-bit digital signal process, TI-TMS320F28379. It is noted that the bilinear transformation is used to realize all control and estimation algorithms mentioned in Section 5 and Section 6.

### 7.1. High-Resolution Position Interpolation

This part verifies the high-resolution position measurement based on the proposed interpolation process in Figure 4. Figure 17 compares two measured position signals, coarse position θ_crs_ and interpolated position θ_intp_, under a 10 Hz sinusoidal position command. In this test, the PM motor is operated without a mirror to clearly demonstrate the position resolution neglecting vibrational load. As seen from the magnified plot, the resolution of coarse position θ_crs_ is around ±0.7 deg (78 μm) without the interpolation. By contrast, the interpolated position θ_intp_ is close to the position command, where the resolution is smaller than 0.01 deg (1 μm). It is concluded that the position interpolation can greatly increase the measurement resolution under the same sensing hardware.

### 7.2. Observer-Based Speed Estimation

This part verifies the observer-based speed estimation in Figure 8. The speed estimation from the direct differentiation ${\hat{\omega }}_{\mathrm{direct}}$ in Equation (8), observer-based standard speed estimation $\hat{\omega }$ in Equation (9), and proposed enhanced estimation ${\hat{\omega }}_{\mathrm{en}}$ in Equation (10) are all compared. The speed is set at 50 Hz rotor frequency under 0.64 g·cm^2^ mirror load. In Figure 18a, if the direct differentiation is applied, a visible filter-reflected phase delay is observed under a 2.5 kHz LPF. Moreover, a small amount of speed variation is resultant during peak speed commands. This variation is caused by the friction torque when the motor mirror rotates across zero position. Comparing to ${\hat{\omega }}_{\mathrm{direct}}$, two observer-based estimated speeds are illustrated in Figure 18b. By adding the torque feedforward in Figure 8, the nearly zero phase delay is achieved on both $\hat{\omega }$ and ${\hat{\omega }}_{\mathrm{en}}$. More importantly, the proposed enhanced estimation ${\hat{\omega }}_{\mathrm{en}}$ results in the lowest speed variation among the three estimated speeds. As a result, better performance is expected if proposed ${\hat{\omega }}_{\mathrm{en}}$ is used for the motion control and acceleration estimation.

### 7.3. Observer-Based Acceleration Estimation

This part verifies the acceleration estimation through the proposed separated acceleration observer in Figure 14. Figure 19 compares time-domain waveforms of acceleration command α*, direct differentiation estimation ${\hat{\alpha }}_{\mathrm{direct}}$, and proposed observer-based estimation $\hat{\alpha }$ in Equation (13). The ${\hat{\alpha }}_{\mathrm{direct}}$ is obtained based on an additional differentiation of ${\hat{\omega }}_{\mathrm{direct}}$ in Equation (8). As seen from Figure 19a, differentiation noises are greatly increased with phase delay under 5 kHz LPF. Moreover, the visible acceleration variation is caused by the speed variation in Figure 18 when the motor moves across zero position. In this case, ${\hat{\alpha }}_{\mathrm{direct}}$ cannot be used for acceleration control. By contrast, the proposed acceleration observer is implemented without the differential controller. As seen from Figure 19b, the differentiation noises are reduced without phase delay. The acceleration variation in proposed $\hat{\alpha }$ is also improved comparing to ${\hat{\alpha }}_{\mathrm{direct}}$. It is concluded that the acceleration feedback control can be implemented using this high-resolution acceleration estimation.

### 7.4. Motion Control Response

This part compares the motion control among different speed feedback signals. In this test, the conventional motion control is compared to the proposed speed/acceleration estimation and control. It is expected that both the transient dynamic and steady-state error can be improved based on the proposed control system.

Figure 20 compares the dynamic operation for the position control with different control and estimation schemes. In this experiment, a step position command is applied to move the mirror from 0 deg to 0.1 deg under 0.64 g·cm^2^ mirror load. In Figure 20a, the conventional motion control in Figure 7 is applied where ${\hat{\omega }}_{\mathrm{direct}}$ in Equation (8) is used for the speed regulation. Although cascaded position P control and speed PI control is designed to remove transient overshoot, the slow 2 ms transient response and visible vibration-reflected position error are resultant. By contrast, in Figure 20b, the transient setting time is improved from 2 ms to 1.27 ms once the observer-based ${\hat{\omega }}_{\mathrm{en}}$ is used for speed regulation. Because of the low-noise speed feedback, a faster dynamic response is achieved under the same controller bandwidth.

More importantly, Figure 20c shows the proposed acceleration control in Figure 13 using the observer-based acceleration estimation. The transient response is further reduced to 0.5 ms. Better control accuracy is also observed compared to (b). In this drive, the bandwidths of position control and speed control are, respectively, at 1.25 kHz and 2.5 kHz. Based on this experiment, it is concluded that the proposed acceleration estimation and control provide the best transient and steady-state performance of the PM motor.

## 8. Comparison between Simulation and Experimental Results

This section makes a comprehensive comparison between simulation and experimental results to analyze the difference between theory and actual implementation. First, the proposed high-resolution position interpolation process is experimentally tested in Section 7.1. Compared to the conceptual interpolation shown in Figure 5a, the experimental results in Figure 17 verify the high-resolution position measurement based on the proposed interpolation process in Figure 4.

Second, the proposed observer-based speed estimation is compared. From the simulation of different speed estimation methods, the enhanced speed observer estimation ${\hat{\omega }}_{\mathrm{en}}$ can achieve a high dynamic estimation response with negligible differentiation noises at high-frequency regions. In the experiment, the bilinear transformation is used to realize the speed estimation algorithm. Comparing with the direct estimated ${\hat{\omega }}_{\mathrm{direct}}$, both two observer-based estimated speeds in Figure 18b have better differentiation noises rejection performance. By adding the torque feedforward in Figure 8, the nearly zero phase delay is observed on both $\hat{\omega }$ and ${\hat{\omega }}_{\mathrm{en}}$. More importantly, the proposed enhanced estimation ${\hat{\omega }}_{\mathrm{en}}$ demonstrates the lowest speed variation among the three estimated speeds without using any low-pass filters.

Third, the observer-based acceleration estimation is compared. Based on the proposed separated acceleration observer, the simulation result is shown in Figure 15. As seen from direct estimated ${\hat{\alpha }}_{\mathrm{direct}}$, considerable differentiation noises appear. By contrast, for enhanced estimation ${\hat{\alpha }}_{\mathrm{en}}$, the 3.5 kHz harmonic information disappears, which cannot be implemented for the high-frequency vibrational harmonic rejection. Based on this simulation, the separated acceleration estimation $\hat{\alpha }$ with another independent observer bandwidth is useful to maintain vibrational harmonic information without differentiation noises. This proposed $\hat{\alpha }$ is suited for acceleration feedback control and disturbance rejection. The experiment of proposed acceleration estimation is illustrated in Section 7.3. Compared to the simulation, the direct estimated acceleration ${\hat{\alpha }}_{\mathrm{direct}}$ needs an additional low-pass filter to eliminate the differentiation noises, leading to the visible phase delay. In this case, ${\hat{\alpha }}_{\mathrm{direct}}$ cannot be used for acceleration control. By contrast, the proposed acceleration observer is implemented without the differential controller. As seen from Figure 19b, the differentiation noises are reduced without phase delay. The acceleration variation in proposed $\hat{\alpha }$ is also improved. It is concluded that the acceleration feedback control can be implemented using this proposed high-resolution acceleration estimation.

Finally, the high-performance motion control through the proposed closed-loop acceleration control is compared. In the simulation, the disturbance rejection is performed by the DRF to analyze the motion control performance. In Figure 12, the corresponding DRF at different frequencies is analyzed. The proposed acceleration control can improve the high-frequency disturbance rejection performance. In Section 7.4, the experiment is used to verify the motion control performance with different control schemes. As seen in Figure 20b with the conventional motion control, the high-frequency disturbance is induced by the mirror load causing the steady-state error and control stability issues. On the other hand, in Figure 20c, the motion control with proposed acceleration control is used to compensate for the high-frequency disturbance. The proposed acceleration control not only improves the transient response but also provides the best steady-state performance. It is concluded that the acceleration feedback control can improve the PM motor control system on both transient and steady-state performance.

## 9. Conclusions

In this paper, a high-resolution acceleration motion control system with separated observers to estimate the speed and acceleration is proposed for motor motion control. Comparing to conventional motion control, the proposed acceleration control improves both the system dynamic response and steady-state accuracy. Based on the model-based observer design, the proposed separated observer not only reduces the differentiation noises but also achieves zero phase delay. By using the proposed interpolated position as the observer input, better acceleration estimation accuracy is achieved. According to the simulation and experimental results, better disturbance rejection performance is demonstrated for the high-performance motion control. Key conclusions are summarized as follows.

A position interpolation is used to increase the position resolution. The high-performance speed and acceleration estimations are implemented with this interpolation process.The proposed speed observer reduces the differentiation noise on speed estimation. A better dynamic response of the PM motor is achieved.A separated acceleration observer is proposed for acceleration estimation. The lowest steady-state error is achieved through acceleration control to suppress the high-frequency mirror vibrational harmonics.

## Figures and Tables

**Figure 1 sensors-22-00725-f001:**
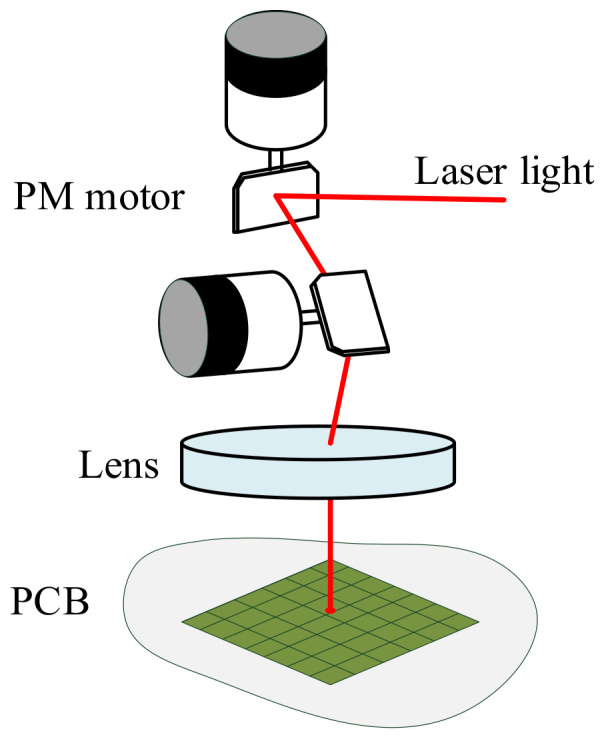
PM motor for PCB hole digging applications.

**Figure 2 sensors-22-00725-f002:**
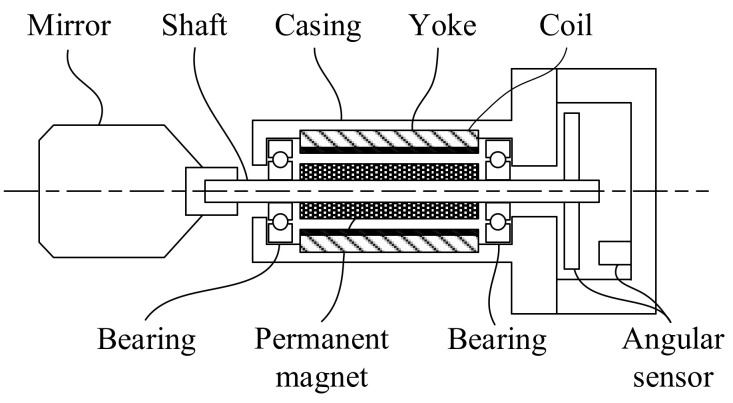
Illustration of analyzed coreless PM motor.

**Figure 3 sensors-22-00725-f003:**
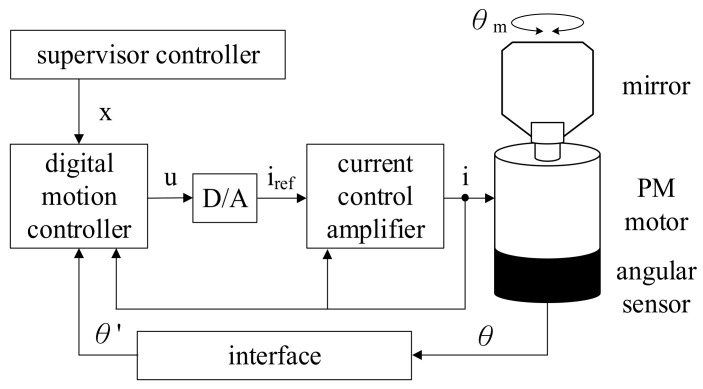
PM motor system.

**Figure 4 sensors-22-00725-f004:**
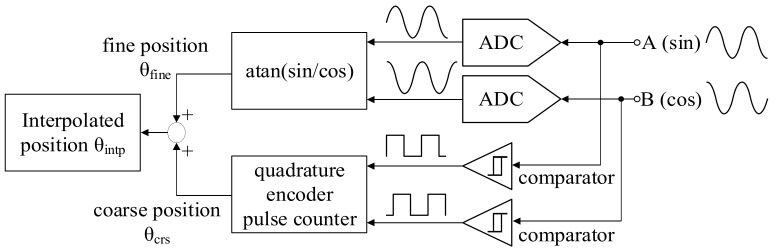
Encoder measurement improvement based on encoder pulse interpolation.

**Figure 5 sensors-22-00725-f005:**
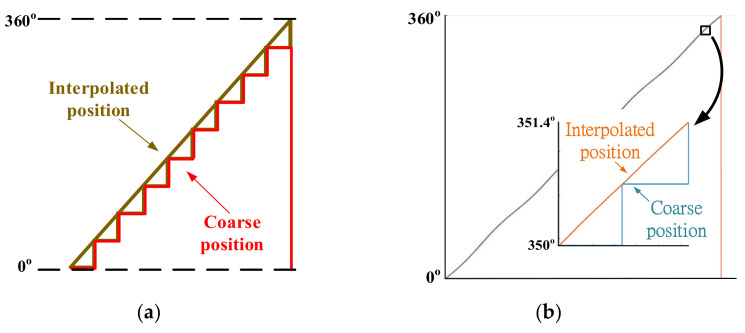
Position interpolation between encoder pulse counts: (**a**) conceptual interpolation process and (**b**) experimental result from the test motor (9-bit pulses + 16-bit ADC).

**Figure 6 sensors-22-00725-f006:**
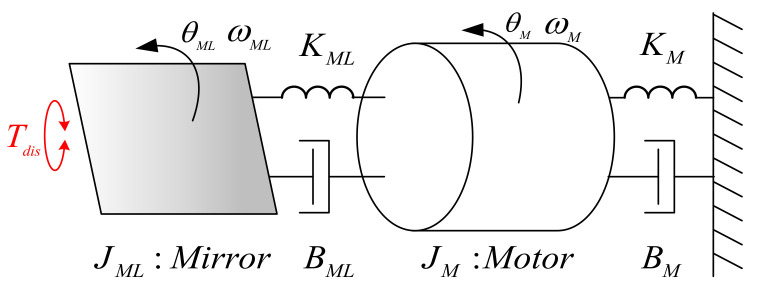
Equivalent model of the PM motor with mirror.

**Figure 7 sensors-22-00725-f007:**
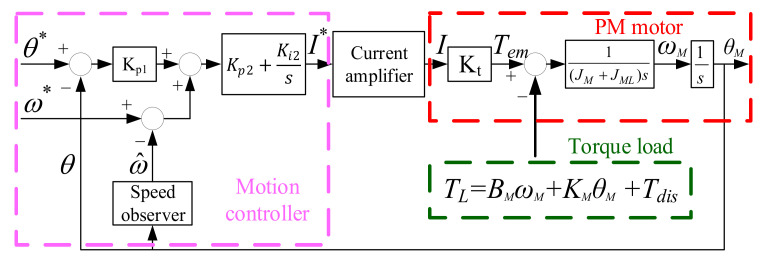
Standard PM motor motion control with position and speed feedback.

**Figure 8 sensors-22-00725-f008:**
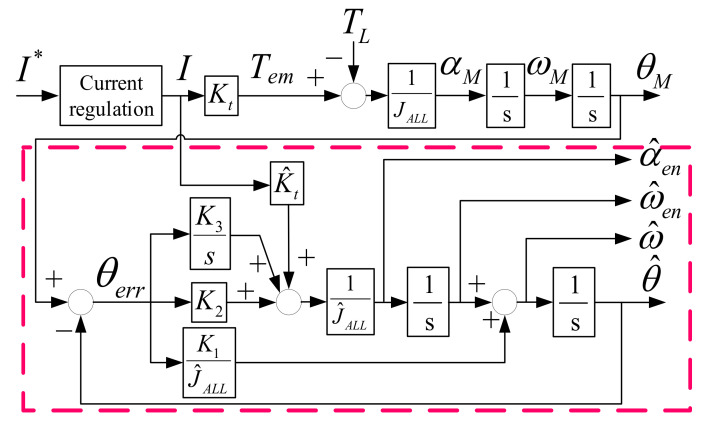
Speed estimation using the proposed observer with torque feedforward for phase delay compensation.

**Figure 9 sensors-22-00725-f009:**
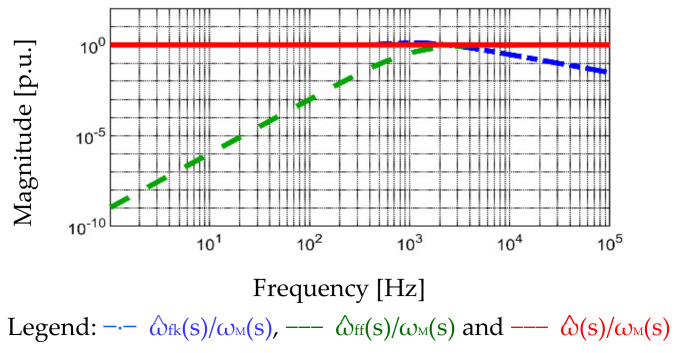
Frequency responses of different speed estimations at different frequencies (ideal parameter estimation, ${\hat{K}}_{t}$ = K_t_ and ${\hat{J}}_{\mathrm{ALL}}$ = J_ALL_).

**Figure 10 sensors-22-00725-f010:**
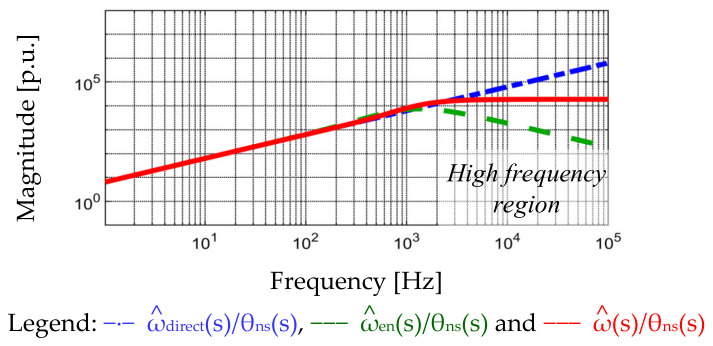
Frequency response of estimation noise attenuation among different speed estimations (ideal parameter estimation, ${\hat{K}}_{t}$ = K_t_ and ${\hat{J}}_{\mathrm{ALL}}$ = J_ALL_).

**Figure 11 sensors-22-00725-f011:**
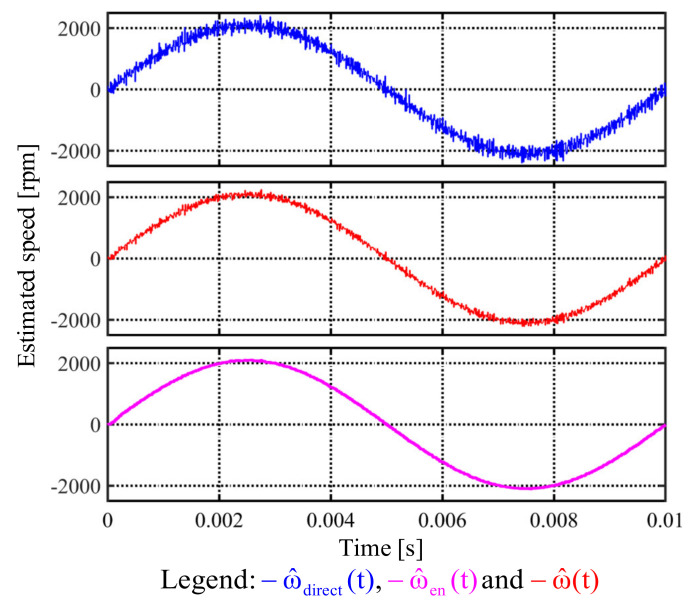
Simulation of different speed estimations under a sinusoidal position feedback signal (0.002% white position noise).

**Figure 12 sensors-22-00725-f012:**
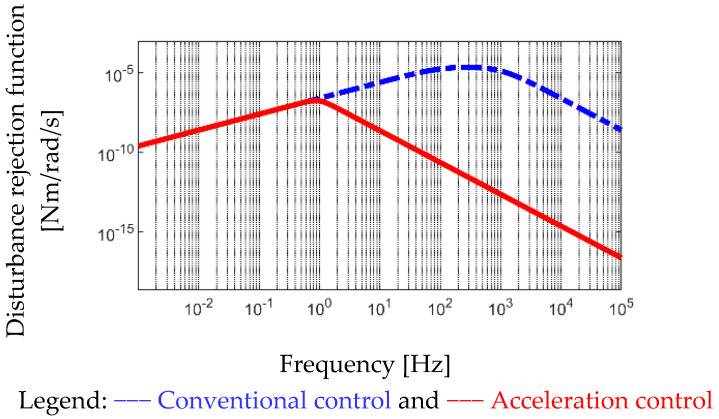
Comparison of disturbance rejection function |θ_L_(s)/T_L_(s)| between conventional control in (11) and acceleration control in (12).

**Figure 13 sensors-22-00725-f013:**
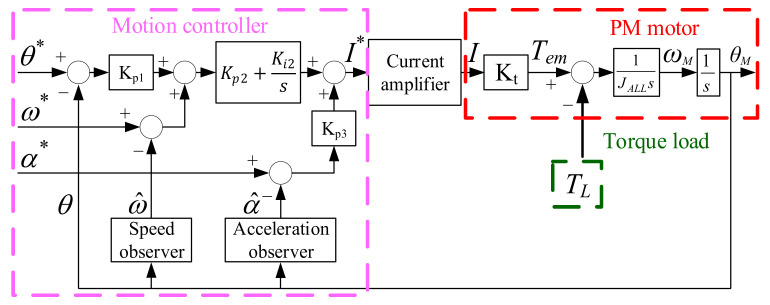
Advanced PM motor motion control with proposed acceleration control to improve the disturbance rejection at high frequency.

**Figure 14 sensors-22-00725-f014:**
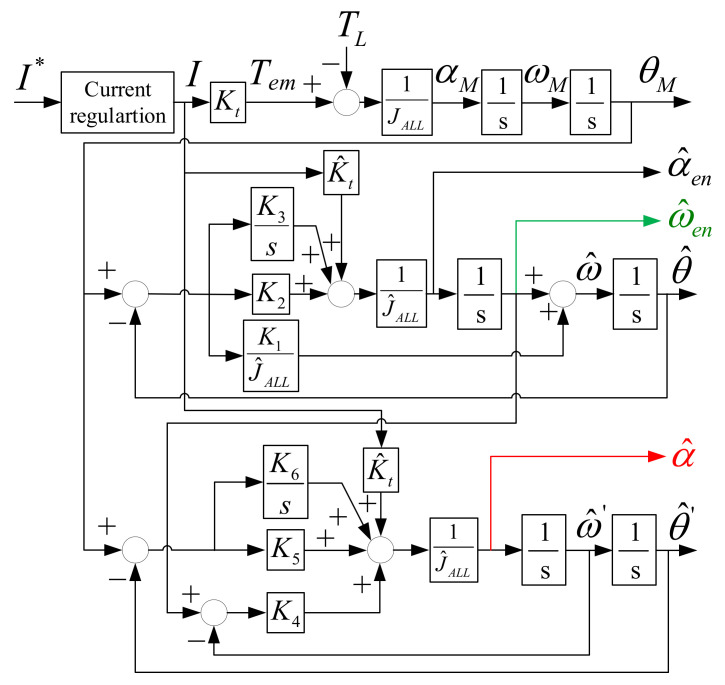
Proposed separated acceleration observer based on the feedbacks of measured position and estimated speed from the prior speed observer in Figure 8.

**Figure 15 sensors-22-00725-f015:**
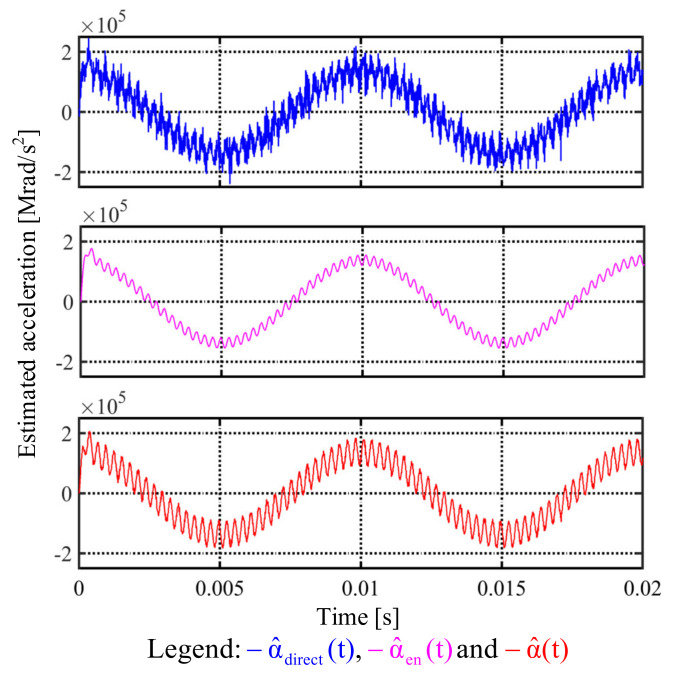
Simulation of different acceleration estimations under a sinusoidal position feedback signal (0.002% white noise).

**Figure 16 sensors-22-00725-f016:**
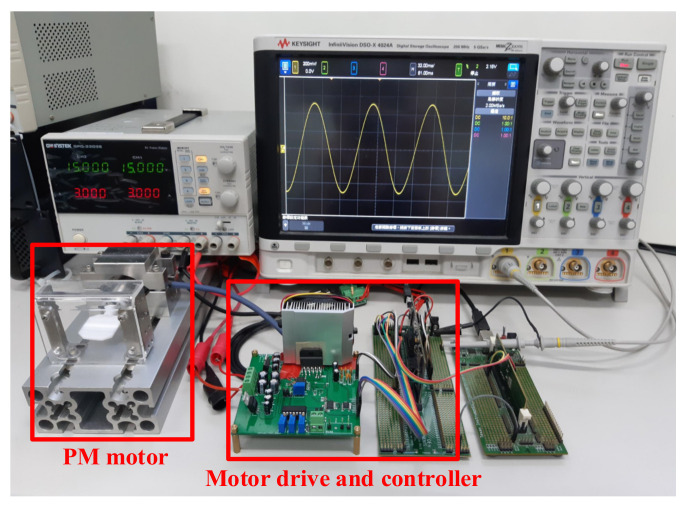
Photograph of PM motor test setup.

**Figure 17 sensors-22-00725-f017:**
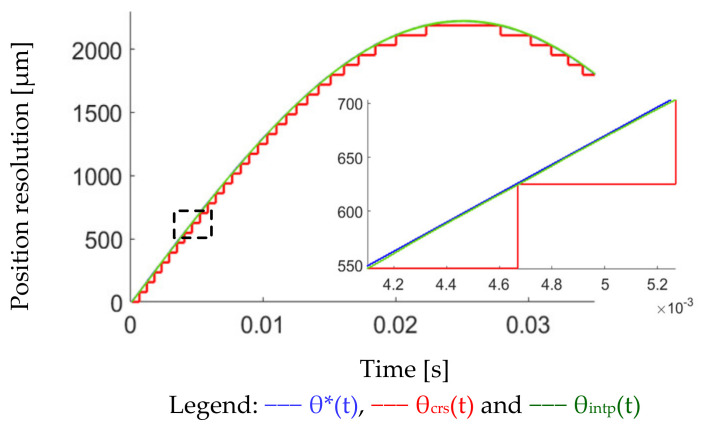
Comparison of different position signals under a sinusoidal position command: position command θ*, coarse position θ_crs_ in Figure 4, and proposed interpolated position θ_intp_.

**Figure 18 sensors-22-00725-f018:**
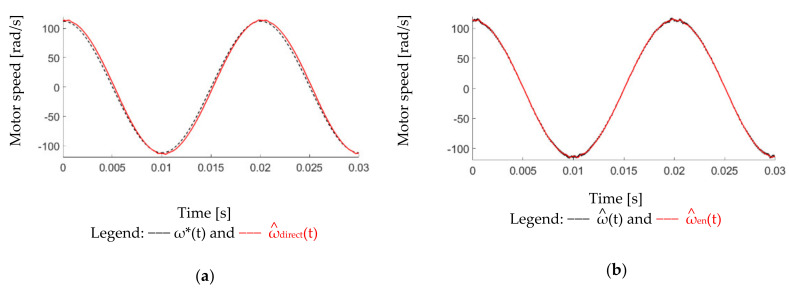
Comparison of different speed estimations under a 50 Hz speed command with 0.64 g·cm^2^ mirror load: (**a**) ω* and ${\hat{\omega }}_{\mathrm{direct}}$ and (**b**) $\hat{\omega }$ and ${\hat{\omega }}_{\mathrm{en}}$ (2.5 kHz speed observer bandwidth).

**Figure 19 sensors-22-00725-f019:**
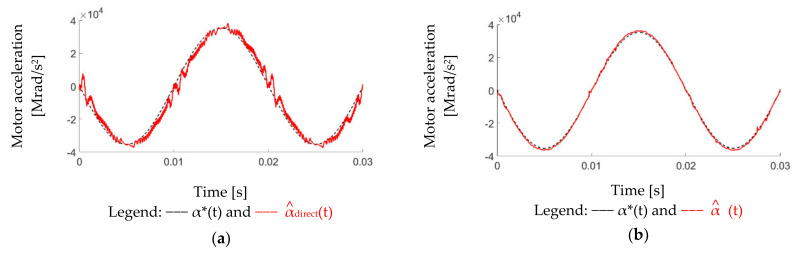
Comparison of different acceleration estimations under a 50 Hz acceleration command with 0.64 g·cm^2^ mirror load): (**a**) α* and ${\hat{\alpha }}_{\mathrm{direct}}$ (**b**) α* and $\hat{\alpha }$ (8.6 kHz acceleration observer bandwidth).

**Figure 20 sensors-22-00725-f020:**
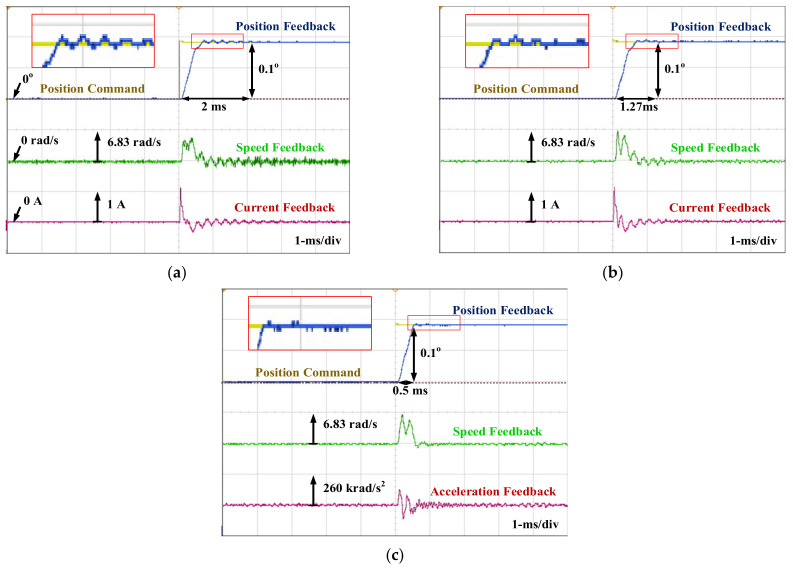
Comparison of position step control performance: (**a**) conventional motion control with direct speed estimation; (**b**) motion control with proposed speed estimation; and (**c**) speed estimation plus proposed acceleration control (0.64 g·cm^2^ mirror load).

**Table 1 sensors-22-00725-t001:** PM motor specifications.

Characteristics	Values
Rotor poles	4-pole
Rated torque	0.0127 Nm
Rated current	1 A
Position rotation	40 deg (maximum)
Resistance	1.7 Ω
Inductance	0.22 mH
Rated voltage	±15 V
Inertia	0.82 g·cm^2^
Sample frequency	100 kHz
Resolution per degree	40 mm/360 deg = 111 μm/deg
Control accuracy (existing drive)	4.41 × 10^−5^ deg
Control accuracy (proposed drive)	1.08 × 10^−5^ deg

## Data Availability

Not available.

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
