# Peer review of "High-Resolution Permanent Magnet Drive Using Separated Observers for Acceleration Estimation and Control†"

_sensors, 2022, doi:10.3390/s22030725_

Round 1

Reviewer 1 Report

The work is interesting and necessary in industrial applications. The structure of the paper and the way the research is conducted are appropriate. The statements made by the authors are justified by the results obtained both using the method of numerical simulation and measurements performed using the experimental model. The method proposed by the authors is superior to those presented in the literature, which is why I consider that the publication of the paper provides researchers with a more accurate solution for controlling the position of laser flux.

My comments on this paper are as follows:

  1. Letters in lines 100-107 should not be bolded.
  2. In the relations (2, 4, 7, 14) some equal signs are missing which are necessary, so they must be completed.
  3. Equations 1,…, 7 are not taken from the literature? If they are taken from the literature, the works from where they are taken must be indicated.
  4. The curves represented using the green color in fig. 11 and fig. 15 are not visible, the coloring must be changed to ensure the visibility of the respective curves.
  5. In equations 2, ..., 10, the size s intervenes, which is not specified what it represents. I think it is necessary for the authors to specify what this variable represents.

Author Response

Please review the attached response letter. Other reviewers comments are also included for the reference. 

Reviewer 2 Report

Improve the control models.

Make a comparison between simulation and experimental results.

Improve the conclusions with referring to simulation and experimental results.

Author Response

(The authors gave the same response as above.)

Reviewer 3 Report

The paper proposes a high-resolution permanent magnet motor drive based on the acceleration estimation and control, mainly meant for high precision laser drilling applications, but usable in wider applications as well.

To achieve the demanded drilling resolution improvements have been proposed in order to o increase the motion control steady-state accuracy balancing transient response. Interpolation of every two encoder counts is proposed to increase the position sensing resolution, where the transient response is improved through the high-resolution position feedback. Also, a closed-loop observer with two independent bandwidths is proposed for
the acceleration estimation.

The paper has good practical output. It gives a nice overview as introduction to the topic, stating what has been made, how the necessary manipulations are usually addressed, and what are the proposed improvements by the authors. The paper cites relevant references on the topic.

The theoretical background is validated by simulations and experimental results. The paper is mainly well written and organized. Easy to read.

Still there are a few improvement points:

Some Figures are hazy and not easy to read. It is worth improving the resolution and quality of those Figs.

Conclusion is very short and given more-or-less as a bullet list. It makes sense to open the conclusion in a few sentences point-to-point. This way the full contribution and usability of the results is more clear to the readers.

Author Response

(The authors gave the same response as above.)
